# Determinants of Disordered Eating Behaviours (DEBs) among Adolescent Female School Students in Riyadh, Saudi Arabia: A Qualitative Study

**DOI:** 10.3390/nu16132119

**Published:** 2024-07-02

**Authors:** Azzah Alsheweir, Elizabeth Goyder, Maha Alzahrani, Samantha J. Caton

**Affiliations:** 1Sheffield Centre for Health & Related Research (SCHARR), School of Medicine & Population Health, University of Sheffield, Sheffield S1 4DA, UK; e.goyder@sheffield.ac.uk (E.G.); or mzahrani@uqu.edu.sa (M.A.); s.caton@sheffield.ac.uk (S.J.C.); 2Department of Community Health Sciences, College of Applied Medical Sciences, King Saud University, Riyadh 145111, Saudi Arabia; 3Health Sciences College at Al-Lith, Umm Al-Qura University, Makkah 21955, Saudi Arabia

**Keywords:** adolescents, disordered eating behaviours, determinants, interview, confidence, body dissatisfaction, body image, Saudi Arabia

## Abstract

The risk of DEB is more prevalent in girls, particularly during adolescence. The onset of DEB can be triggered by many inter-related factors, including biological, social, parental, and psychosocial. To date, very little is known about the determinants of DEBs in Saudi adolescent girls. Using a qualitative analysis, this study explored potential determinants of DEB among adolescent girls in Riyadh. Eighteen semi-structured interviews were carried out with adolescent girls (mean age = 14.06, SD = 0.87) who reported a high risk of DEB (EAT-26 ≥ 20) in intermediate and secondary schools in Riyadh. The mean weight was 51 kg (SD = 11.8) with BMI ranging from 14.18 kg/m^2^ to 27.51 kg/m^2^. EAT-26 scores ranged from 21 to 42 (M = 26.8, SD = 5.6). Data were transcribed and revised, then themes and sub-themes were assigned using MAXQDA 24. The most common DEBs reported were dieting and binging, followed by induced vomiting. Major themes were related to negative cognitions, conscious imitation/copying behaviours, bullying, comparisons, and negative comments. Some participants identified the possibility of biological and familial factors in increasing the likelihood of DEB. Our findings provide a framework that could be used to increase understanding of DEB and inform the development of interventions to address underlying causes of DEB in Saudi adolescent girls.

## 1. Introduction

Disordered eating behaviours (DEBs) are ranked as a highly prevalent public health concern related to body dissatisfaction and weight control in adolescents [1]. The literature indicates that disordered eating (DE) is prevalent across various racial, ethnic, cultural, and socioeconomic backgrounds with the highest proportion occurring in adolescents, particularly in mid to late adolescence [2]. These behaviours include dieting and food restriction, binge eating, induced vomiting, consuming laxatives and diet pills, and intensive exercising. These behaviours are considered maladaptive behaviours, yet they seem to be goal directed toward fulfilling intrapersonal (e.g., regulating emotions) or interpersonal (e.g., seeking attention or care) purposes [3]. Girls are reported to be more vulnerable to DE as they are more likely than boys to be preoccupied with body image and physical appearance [4]. Since normal body weight lies on a continuum, pressure to reach the alleged perfection can elicit emotions of distress [5].

Several determinants have been associated with DE in adolescents, including biological, social, parental, and psychosocial factors [6]. Research suggests that growth spurt and rapid changes in body shape and size may play a significant role in DEB, as adolescents may have low self-esteem and negative body image due to growth variations [7]. Graber (2013) clarified that girls who show early signs of maturation are more likely to experience bullying and teasing regarding their changing bodies. Consequently, the likelihood of developing DEB is increased [8].

Furthermore, negative emotions and poor mental health (e.g., depression, anxiety, stress, poor self-esteem, etc.) are associated with DE [9]. For example, Hafsa (2014) reported that adolescent girls (15%) in Riyadh are engaging in binge eating episodes to reduce feelings of stress and depression [10]. Another study targeted 471 high school students in Makkah to explore the risk of DEBs. A key finding indicated that high levels of stress are significantly associated with an increased risk of DE [11].

Despite individual differences in thinking and emotional regulation, some personality traits tend to be associated with DEB, like perfectionism and cautiousness [12]. Childhood personality traits, particularly perfectionism and rigidity, increase vulnerability to such disorders [13]. A review focusing on personality traits and ED confirmed that personality disorders, such as perfectionism, obsessive compulsiveness, and negative emotionality, are common among people diagnosed with ED [14].

Shape concerns, negative body image, and the preference of a thin body ideal are considered significant determinants of DEB [15]. A growing proportion of adolescent girls feel overweight or fat and have a strong desire to lose weight. These feelings may trigger adopting the “dieting mentality” and increase vulnerability to DEB [16]. A longitudinal study focused on understanding the aetiology of body dissatisfaction in adolescent girls aged 14 to 18 years concluded that their body dissatisfaction is linked to the desire to be thinner, weight control practises, negative comments about weight, and BMI [17]. However, in our recent study involving adolescent girls aged 12 to 19 years attending intermediate and secondary schools in Riyadh, BMI did not appear as a significant predictor of DEB practises [18]. This does not necessarily indicate that adolescent girls with obesity face the same risk as girls who are underweight. It should be noted that the risk is not directly related to BMI and weight status, yet it is affected by weight status perception and body dissatisfaction. A study on European adolescent girls explored body perceptions and weight status and confirmed that girls with BMI ≤ 50th percentile tend to overestimate their weights and struggle with feeling unattractive [19].

The prevalence of DE is strongly linked to familial and parental factors [20,21]. Parents’ comments on body, weight, and physical appearance have a substantial influence on eating patterns and body concerns, emphasising that both mothers and fathers are significant sources of influence on their children [21]. Coercive parental control and critical family environment are precursors to the development of DEBs [22]. Similarly, peers and friends can strongly impact adolescents’ engagement in risky or harmful behaviours [23]. Adolescents and their peers have comparable perceptions of weight and shape, which are primarily caused by peer influence [24].

Vulnerable adolescents are persuaded by the media to adopt negative attitudes and behaviours related to their bodies [25]. Recent evidence suggests that media pressure has the greatest influence on body dissatisfaction among female students, followed by peer and parental pressure [26]. A study examining the role of social media on eating behaviours and body concerns among 681 adolescents confirmed that using social media is associated with body dissatisfaction, DE, and muscle-building activities [27].

Within the last decades, the Kingdom of Saudi Arabia has experienced a marked transition in dietary habits and food choices due to economic accelerations and sociocultural changes in the Arabian region. From 2003 to 2013, Saudi Arabia experienced significant modernization, which improved the wealth and wellbeing of the society [28]. In 2014, Saudi Arabia was ranked to be the world’s 19th highest economy and classified as one of the leading economic countries. Household income increased noticeably by 75%, owing to the improved public employment sector and wages. The degree of urbanisation is elevated compared to other European countries, with almost 83% of Saudi society living in cities [28]. These changes resulted in negative modifications to diet and food choices with dietary patterns being adjusted drastically to mimic Westernised behaviours, which are significantly associated with elevated rates of obesity [29]. The thin body ideal is one of the significant Western concepts that continues to diffuse in society, replacing the favourable full body from Arabic culture [4,30]. With rapid cultural shifts to emulate Western values in weight and shape, Saudi adolescents have more vulnerability to eating pathology, body image, and psychological conditions [31].

Even though research has explored potential determinants of DEBs, findings are contradictory and require further investigation [32]. The nature of these behaviours and their prevalence are not clearly explored in Saudi Arabia. Our recent study on adolescent girls attending intermediate and secondary schools in Riyadh indicated that about 30% of students reported a high risk of DEB [18]. Overall, there is a paucity of research that has examined factors that might contribute to the development of DEB in Saudi adolescents. Few studies have particularly examined key factors such as socioeconomic status, family size, parental education, and occupation [33,34].

To date, no study has explored DE in the Saudi female adolescent population within a qualitative context. Therefore, more formative research that promotes understanding the equivocal interplay of factors contributing to DEB is required, by closely examining female students’ experiences with DE to identify contributors and factors triggering such behaviours. This study provides preliminary insights into the potential causal inferences associated with DEBs. A general exploratory scheme is applied to identify the common determinants and factors causing the development of DEBs. The objective of this study was to take an initial step toward addressing potential determinants and predictors of DEBs among female adolescents in the city of Riyadh, Saudi Arabia.

## 2. Materials and Methods

### 2.1. Theoretical Framework

Determinants and predictors in mental health are linked to various factors (e.g., individual and social) and their interactions. Therefore, it is prudent to conceptualise the underlying biological, psychological, and sociocultural perspectives associated with mental conditions [35,36]. According to the WHO (2012), targeting different multi-layered factors is important to reduce illness and improve mental health [37].

In this study, the theoretical framework for exploring determinants and predictors of DEBs was formulated by Bronfenbrenner’s bioecological model [38,39,40]. Based on concepts derived from psychology and social ecology, it is necessary to comprehend an individual’s behaviour and psychological development within their social environment [41]. The bioecological model developed by Bronfenbrenner focuses on the social environment addressing the interaction among intrapersonal characteristics, interpersonal relationships, environmental context, and time with their role in shaping and modifying an individual’s attitudes and behaviours [41].

Bronfenbrenner’s bioecological model is designed according to four interconnected systems presented in concentric circles. The developing individual is located at centre of the diagram surrounded by inter-related systems. The inner circle (microsystem) classifies direct and immediate relations with family, friends, and objects. The inner circles (mesosystem and exosystem) describe several environments experienced by the developing individual as they interact with and influence the microsystem. The outer circle (macrosystem) is identified by cultural values and dominant norms. The immediate ring surrounding the individual refers to the techno-subsystem, which indicates technology and the use of devices (phones, computers, social media, etc.) and their capacity to shift many direct interactions, as they act as mediators in an individual’s interactions with others [42,43].

### 2.2. Study Design and Setting

This qualitative study focused on interviewing adolescent female students who reported a high risk of DEB. Participants, from intermediate and secondary governmental schools in the city of Riyadh, were selected from a previous study [18]. Data were collected between December 2023 and January 2024. Ethical approval for this project was granted by the Research Ethics Committee at the University of Sheffield (Reference Number 050493) and the Research Ethics Committee at King Saud University in Riyadh (No. E-23-7551), following the principles articulated in the 1975 Declaration of Helsinki.

### 2.3. Participants and Recruitment

In the early quantitative study, four governmental intermediate and secondary schools were selected and 416 female students were recruited to measure the prevalence of DEB by using the Arabic version of the Eating Attitudes Test (EAT-26) [33,44,45,46,47,48]. Students who reported a high score on the EAT-26 test (score ≥ 20), classified as high risk of DEB, were interviewed. The ages of the students ranged from 13 to 16 years, who were recruited from two schools in Riyadh, the 128 Secondary School (North) and the 213 Intermediate School (North) in AsSahafa district.

### 2.4. Procedure

Following approval from the Ministry of Education (MoE) in Riyadh to visit the designated schools and interview students, school principals were contacted to distribute the participant information sheets (PISs) and parental consent forms to identified potential participants. With permission from the class instructor, students who returned the signed consent form were contacted and an interview time was arranged. Interviews were carried out in the school environment. At the beginning of the interview, each student received a student PIS and an informed consent form. The PIS, which comprised information on the study, the process of the interview, and confidentiality, was explained and the student was asked to sign the informed consent before starting the interview and the audio recording.

### 2.5. Interviews

In this study, semi-structured interviews were carried out. The duration of the interviews ranged from 30 to 40 min. The interview guide questions were developed by applying Kallio’s et al. (2016) five phase framework for semi-structured interviews [49]. As a first phase, semi-structured interviews were identified as a thorough and suitable approach to gathering data on the DEBs. In the second phase, prior knowledge was comprehensively retrieved and understood, addressing the need for empirical research on DEBs among Saudi adolescents. In the third phase, ten preliminary questions were formulated for the interview guide considering the EAT-26 questions and applying resources aimed at assessing adolescents with eating disorders [50,51]. The fourth phase included pilot testing the interview guide through internal testing and field testing. Three internal testing meetings and two field tests were implemented to evaluate interview questions. Internal testing meetings were held by the principal investigator (PI) (A.A.) and two reviewers (S.J.C. and E.G.) to examine interview questions and make needed adjustments. Field tests were conducted by a PI (A.A.) with five Saudi adolescents aged 12 and 13 years. In the fifth phase, a list of questions was presented in the complete semi-structured interview guide (Appendix A). The final guide was then translated to Arabic by the PI and used for the interviews.

### 2.6. Data Analysis

Data were analysed using Braun and Clarke’s thematic analysis guide applying Bronfenbrenner’s bioecological model as a theoretical framework [40,52]. Arabic responses were transcribed verbatim by the PI (A.A.) with the involvement of a reviewer (M.A.) who is an Arabic native speaker, revising 10% of Arabic transcripts. Themes were generated independently and the final themes and sub-themes were reached by consensus (A.A. and M.A.). Quotes, themes and sub-themes were then translated into English and reviewed by two reviewers (S.J.C. and E.G.). MAXQDA 24 software was used for retrieving responses and coding themes. Themes and sub-themes were subsequently classified according to Bronfenbrenner’s bioecological model systems.

## 3. Results

### 3.1. Sample

Eighteen interviews were conducted in total, two students aged 16 years from the 128 Secondary School took part, and 16 students aged 13–15 years from the 213 Intermediate School in Riyadh (mean age = 14.06, SD = 0.87). Height (Ht) ranged from 144 cm to 165 cm (mean Ht = 155.2 cm, SD = 5.7 cm) and weight (Wt) ranged from 31.9 kg to 69 kg (mean Wt = 51 kg, SD = 11.78 kg). Participants’ BMI ranged from 14.18 kg/m^2^ to 27.51 kg/m^2^ with a mean of 21.03 kg/m^2^ (SD = 3.99 kg/m^2^). EAT-26 scores varied among participants, ranging from 21 to 42 (mean = 26.78, SD = 5.59).

### 3.2. Common DEB

Adolescent girls described the most prevalent behaviours as personally experienced or observed in other girls (sisters, friends, or relatives). Dieting and binging to control weight/shape were reported as the most prevalent behaviours, followed by induced vomiting. Using laxatives or diuretics and engaging in intensive exercises were the least behaviours reported.


*“Like the one who deprives herself of food all day long, and when the night comes, she eats… and literally eats twice as much as she can eat in the morning”.*
*Participant* *2*


*“There are girls like that in school, and sometimes they never eat. They say, “I want to lose weight.” I mean… and they say that “I only eat a little bit”.*
*Participant* *5*


*“Every time she eats, she goes to the bathroom and deliberately vomits…. She does not like this thing, and she is trying to stop, but that’s it! since she started doing this thing, she continues because she doesn’t want to reach… She doesn’t want to gain more weight”. *
*Participant* *17*

Adolescent girls explained dieting as fasting or a state of food deprivation. Some even discussed that they might eat once a day as they lack the appetite to eat or they feel that there is no need to eat.


*“My stomach is telling me “don’t eat”… I don’t feel like eating… It seems like there is something inside but there is nothing… The feeling of nausea…the pain… I get pain from here to there (she points from the throat to the stomach)…and heat…I don’t want to eat…I don’t want to taste anything”*
*Participant* *4*

Many girls reported experiencing dizziness and fainting due to food avoidance and lack of nourishment. Girls who reported dieting indicated that they engage in binging episodes by the end of the day after depriving themselves of food for the entire day.


*“I fainted, maybe two or three times, I swear I don’t remember…One time I fainted in my room and no one knew”.*
*Participant* *7*


*“I stopped eating, but I didn’t get any binging episodes. I stopped for a long time and continued doing that until I became dizzy every time I went to school”. *
*Participant* *11*

Girls described that vomiting could be induced after a binging episode to reduce feelings of shame and guilt. Girls who engaged in vomiting explained that induced vomiting is an easier way to control weight and suppress appetite.


*“But if my family forced me to eat something, I would vomit after it so that they would make me… I mean, they would see me vomiting… No, you won’t eat anything anymore……Okay, you want to vomit, then I will give it to you and let you vomit. After that, I started vomiting without anything… without deliberately doing it. I vomit… I let myself vomit”. *
*Participant* *7*

Some students mentioned intensive exercising as a method for burning calories and losing weight.


*“My cousin… I remember one time she was dizzy and they took her to the hospital because she had done exercising… and had not eaten anything all day… I told her, “Why did you do that?” She said, “I want to lose weight”. *
*Participant* *3*

Similarly, diet pills were also not common; however, many students claimed that they were considering taking appetite-suppressant pills or injections to control their weight.


*“I mean, more than one girl talked about injections that suppress the appetite and said that they should go talk to a doctor and see if they can use them or not”. *
*Participant* *12*

Concerning the frequency of engaging in DEB with age, responses differed among participants. Most adolescent girls believed that DEB would be reduced with age, with some students linking this reduction to comprehension and understanding. They clarified that by growing older they will come to understand the complexities of these behaviours and perceive their consequences clearly and accurately. A few students thought that DEBs would increase with age as they anticipated growing older to be associated with aggravated feelings of desiring to be thin and attractive. However, some students were not sure if DEBs would increase or decrease, as they stated that it is dependent on multiple underlying factors (current weight, confidence level, family’s motivation and acceptance, etc.).

### 3.3. Views on Weight and Shape

In terms of personal weight and shape, the majority of adolescent girls indicated that they are not happy and content with their weight/shape, with the majority of adolescent girls stating that they need to lose or gain weight. Height was also mentioned; a few girls specified their height and described that they would prefer to have a taller body and continue to wear platform trainers to appear taller.


*“I suffer from my body a lot, with clothes and how I see myself, and in terms of what people say to me… so I suffer from obesity.” *
*Participant* *12*


*“Yes, when I was in my first year of intermediate school, I used to say that I wanted to be thinner, and until now, I want to be” *
*Participant* *16*

In terms of others’ weight and shape, a few students believed that girls their age like their weight and are comfortable with their shape, while most students thought girls their age are not content and want to be slimmer. Some students stated that girls are afraid of their weight even if they have a normal weight. They described that girls are afraid of standing on the scale as they constantly have this desire to reach the “perfect body”. They defined the perfect body as the ideal weight, sculpted waist and attractive height.


*“They see this slim, tall, sculpted body…is the perfect body that every woman dreams of” *
*Participant* *2*


*“I mean most girls now… are afraid of their weight… I mean… they are afraid that someone will see their weight… I see them getting nervous and afraid when they are asked about their weight” *
*Participant* *5*

### 3.4. Determinants of DEB

Adolescent girls discussed the main determinants and factors causing DEBs. Analysing responses generated six main themes and 12 sub-themes (Table 1). Below is a description of each theme/sub-theme and designated bioecological model system, together with pertinent illustrative quotes retrieved from adolescent girls’ responses.

Negative Cognitions

Participants explained that negative cognitions and thoughts would be significant in causing DEBs. Overthinking was defined by girls as a pathway for developing negative thoughts and cognitions.


*“I’m afraid that I’ll start thinking… I’m thinking again and more thoughts will go on and on” *
*Participant* *7*


*“Overthinking for sure…I mean, I feel like if you don’t occupy your time, you’re definitely thinking about anything that it’s not good, so I don’t know… I think that” *
*Participant* *18*

1a. Self-esteem and confidence issues

Participants defined negative cognitions as a core factor distorting their level of confidence. They see their bodies as different or strange compared to a slim body, which aggravated feelings of shame, self-doubt, and low confidence.


*“Confidence… no confidence… I want to be like normal girls and become a beautiful girl with confidence…” *
*Participant* *4*

Negative self-talk is another aspect highlighted by adolescent girls that can significantly impair their confidence level. Negative self-talk is related to appearance/shape and worn clothes. Many students expressed that although they desire to wear tight clothing and show off their bodies, they feel uncomfortable doing so since they lack confidence.


*“She is trying to bring down her confidence level by saying I feel that I do not look nice… and today I feel that my clothes are not nice… It is like my clothes aren’t suitable for my body… This clearly shows that she is not confident”. *
*Participant* *17*

1b. Negative Body Image

Girls linked negative body image with clothing. They expressed their feelings when they try new clothes or buy large sizes. They feel embarrassed about how big they are and ashamed of always buying and trying large sizes. They consider their bodies as abnormal and are tired of wearing large sizes.


*“If I try clothes on… I look at myself… I mean I say I don’t… That’s it, I don’t want to eat anymore”. *
*Participant* *4*


*“I see girls in school… and I watch them… I see them… I mean, they are all skinny and they wear things like that, but… I don’t know… I feel that I want to be skinny, I don’t want this body”. *
*Participant* *11*

Even if girls are thin, they have a problem with negative body image and want to lose more weight. They are afraid of their weight and they avoid standing on the scale.


*“I see that they have normal bodies, but even though they are ashamed of this thing… It doesn’t matter if she is fat or thin… In general, she is ashamed of her body and makes sure that no one is behind her to see the scale”. *
*Participant* *17*

2.Conscious Imitation/Copying Behaviours

Adolescent girls expressed that when looking at other people and their bodies, they want to imitate them and copy their behaviours either in real-life or on social media.


*“It is a trend and they want to go with it… You see people, I mean… a group of people is doing this behaviour and you are going to do the same…” *
*Participant* *12*

2a. Real-life Models

Girls discussed that when they notice a girl with a beautiful body they want to be like her and imitate her behaviours. They explained that they wanted to be admired and complimented for their looks like her. Some of the participants indicated that they are ignored and unseen by others (e.g., family, friends, and school peers), but they will be noticed when they become a copy of any attractive girl.


*“I don’t hear anything positive about me and no one is complimenting me… so I will go and become like her… lose weight… I do and do and do… just to become a copy and paste of her”. *
*Participant* *2*


*“They imitate each other… Just like one of my friends, she is copying my other friend… She wants to be like her, you know” *
*Participant* *8*


*“But there are pretty girls, so it’s normal for you to say that she is pretty… And how can I be like her”? *
*Participant* *16*

2b. Social Media—TikTok and Instagram

Girls identified the role of social media and conveyed messages of the ideal body and perfect life. They explained that with social media’s consistent focus on the slim and sculpted body, they doubt their appearance and weight until believing that they are neither beautiful nor attractive. They live under persistent psychological pressure to lose weight and become prettier by imitating what they see on screens.


*“Girls can see things that affect them… like, for example, they look at magazines… or they can watch TV… they see bodies… they say her body is beautiful… or she looks beautiful… I want to change myself or I want to be like her… She could say “Oh wow, I want a beautiful body, sculpted waist and attractive height” *
*Participant* *3*


*“Okay, this one may have a passion for getting slimmer, but then it becomes her obsession all the time is just like… like the manic stage, that she can’t stop thinking about her body…” *
*Participant* *7*


*“I mean, anything that comes out as a new trend, they don’t care if it’s good or not good… Any trend that comes out on TikTok… they will copy it right away, they don’t care if it’s good or not good… If it’s harmful or not harmful”. *
*Participant* *12*

Most adolescent girls mentioned the influence of Korean celebrities and K-pop. They explained that they are imitating their diets and exercises to have skinny and attractive bodies like them, despite how challenging it is to stick to diets that limit their intake of calories and other nutrients.


*“The K-pop diets… So wake up in the morning and eat half an apple, then wait until the afternoon and eat half a banana and two nuts… Don’t eat lunch or rice…Cut out something called rice”. *
*Participant* *16*


*“In intermediate school, we had the Korean girls trend… You know… We want to be like them and they are soo skinny… They have different kinds of diets, but they are so so so strict and they exercise all the time…” *
*Participant* *18*

3.Bullying

Adolescent girls identified the negative effects of bullying on weight and appearance in real-life or social media (cyberbullying). Bullying is a major issue that adds more pressure to be slimmer and lose weight.


*“Of course, she feels pressured and sad from the bullying that happened to her… The more the pressure, the greater the desire to go on a more restricted diet” *
*Participant* *1*

3a. Traditional bullying—Siblings and School Peers

Girls identified traditional bullying as deliberate and continuous acts of verbal aggression from siblings at home or other peers in school. They confirmed that it can severely impact their mental condition and academic achievement. Many participants explained that bullying based on weight and shape will eventually lead to depression, isolation, lack of school punctuality, and persistent absence.


*“Girls like to bully… when they see a fat girl… they join on her… they gather around her…for no reason…and then they shout at her and raise their voices and laugh…they say you are fat…” *
*Participant* *3*


*“If you are bullying me then that’s it! She no longer wants to talk to anyone, she becomes isolated… She leaves her relationships… She leaves everything… She is afraid… She is afraid that she will hear more harsh words… She starts crying in school, crying at home, crying in the bathroom…” *
*Participant* *7*

3b. Cyberbullying

The girls reported that cyberbullying concerning weight and appearance that occurs through social media and apps causes more harm than traditional methods of bullying. These actions are identified on social platforms by posting negative/abusive comments on weight, shape, or appearance by anonymous and unknown accounts.


*“She was fake, so I did not know… She used to send messages and I was afraid… I was afraid of the whole society… I no longer wanted to be close to anyone… I was afraid to trust anyone and then they will do the same, so I was afraid… I would go back. I would not talk to anyone…” *
*Participant* *7*


*“For sure she is going through a state of depression, but she does not know who is this person insulting her… She can’t confront him or talk to him”. *
*Participant* *17*

Many girls mentioned a popular app used by students called Tell, which allows enrolled students at a certain school to text each other about homework, assignments, and tests. They explained that it is similar to Instagram; however, it lacks the feature of posting photos. Participants have encountered many incidents of cyberbullying through sending aggressive and abusive texts related to weight and shape from anonymous accounts that are visible to other students in the school.


*“It’s on Tell, someone from school… So I thought I would always come to school like this! So, who is she?! You know I deleted it because I couldn’t handle that anymore, I was frustrated and tired… so tired of it” *
*Participant* *18*

4.Comparisons and Negative comments

Adolescent girls explained that comparisons and negative comments can be destructive, leading to an increasing the risk of DEB. As it may occur from the immediate surrounding environment including family and relatives (interpersonal), it could also originate within the individual (intrapersonal).


*“Comparison takes a person to a stage that no one can imagine except this person. Comparison is never a good thing”. *
*Participant* *9*

4a. Intrapersonal Comparisons

Adolescents described how they struggle with self-comparisons in relation to their weight perceptions and body image. Comparisons stem from feelings of shame, jealousy, and a strong desire to be slimmer.


*“Because I see all the people now… Why am I the only one who is fat? I compare myself with these people… Personality doesn’t matter to me… Personality doesn’t matter to me as much as I care about appearance.” *
*Participant* *16*


*“She sees someone whose body is thin, for example, and she is a little fat… so she starts comparing, why is she thinner than me and I am fatter than her”? *
*Participant* *17*

4b. Interpersonal comparisons and comments

Comparisons from family, particularly parents, were identified by the girls as the most intense source followed by siblings and relatives. Most girls even agreed that comments from the mother would hurt more than the father or siblings. Comparisons could be made with other people or a thinner or younger version of oneself.


*“Like when my mother says, “Be like your sister… like the comparison… I had a friend before, and her mother used to compare her to the school girls… the school girls!” You don’t know them… She didn’t care about herself and was not well” *
*Participant* *2*


*“My relatives used to compare a cousin of mine with a girl… She was really affected by this thing… she stopped eating and tried to lose weight… They are used to telling her” Look at her, she is your age… Look at her weight and how she is taking care of herself and her shape” *
*Participant* *9*


*“Even my family got upset and said that they saw how I was skinny before, and now they see how I have a strong appetite, and how I can’t stop eating” *
*Participant* *12*

5.Biological determinants

Some adolescent girls described the role of biological determinants including puberty/hormonal changes and fear of chronic disease as factors causing DEBs.

5a. Puberty and hormonal changes

The girls explained that during puberty appearance and body shape change rapidly, making it difficult to accept and embrace these changes. At the same time, they expressed being temperamental and sensitive with hormonal shifts.


*“I feel that at this age you become extremely sensitive… especially at the beginning of puberty… you become sensitive… My personality was not like that at all when I was young… I had a different personality… I don’t know… but it is my teenage period”. *
*Participant* *8*


*“I feel that with intermediate years, all this age is difficult with puberty… In intermediate years the entire appearance changes so they won’t be willing to fully accept these changes” *
*Participant* *17*

5b. Fear of chronic diseases

Few adolescent students mentioned that DEBs could be practised due to fear of developing diabetes or other serious conditions. They see weight gain and obesity as a danger stage, which is linked to many diseases.


*“They are afraid that they will become diagnosed with a serious disease… I mean, for example, they are afraid that they will get… a lot of these things. They are afraid from eating too much unhealthy food… Maybe… they will get cancer… They will get diseases. So they try so hard”.*
*Participant* *1*


*“Diseases… A cousin of mine used to eating all the time and she said it’s OK, suddenly she got diabetes… This thing scared me… because I used to eat a lot every once in a while, so I started not… eating, and I started telling my friends…” *
*Participant* *5*

6.Family-related Determinants

Some students discussed the role of family-related determinants in engaging in DEBs. They considered the nature of the relationships among family members and the environment of the house as the main factors interacting with DEBs. They also mentioned that DEBs could be a method of attracting attention when they feel neglected or ignored.

6a. Family situation/house environment

Girls identified problems that occur between family members, especially parents, as major factors leading to DEBs. Living in a negative environment and experiencing aggressive and stressful conditions can generate feelings of sadness and insecurity, which eventually lead to DEBs.


*“I can’t say that all the mothers and fathers understand each other and so on… I mean, their problems… I mean, the problems that happen between them still affect the family”. *
*Participant* *15*


*“Problems in general… The girl comes with zero appetite, she didn’t eat for two days because of the problems happened at home… I mean the screaming and talking you hear from them is… enough… I don’t want to eat… or when are sitting on the table for lunch, and they start arguing… everyone gets up” *
*Participant* *16*

6b. Care and attention

Some adolescents defined practising DEBs as a way of getting attention and being seen. They explained that DEB would have an impact on their appearance and overall health, making them look pale and poorly, which would motivate parents to nurture them and provide additional care and attention.


*“If you become tired and sit alone… sad and depressed… and your weight goes down… your family says what’s wrong with you… they start treating me kindly and stop pressuring me” *
*Participant* *2*


*“She attracts attention… I feel that at this age some mothers treat their daughters as if they were adults” *
*Participant* *8*


*“Some girls do that, because their families neglect them, so they do this…they want them to care more” *
*Participant* *10*

## 4. Discussion

This qualitative design aimed to explore the determinants of DEB, according to Brofenbrenner’s bioecological model, by interviewing Saudi female adolescents attending intermediate and secondary schools in the city of Riyadh, Saudi Arabia. The process in this study (selecting schools and participants) was based on a quantitative study implemented previously using the EAT-26 test [18]. Adolescent girls indicated that dieting, binging, and induced vomiting were the most common behaviours practised. A key finding of this study was that negative cognitions, conscious imitation/copying behaviours, bullying, comparisons, and negative comments were identified as primary determinants of DEBs. Other determinants that could increase the likelihood of DEBs were biological, including puberty and hormonal changes and fear of chronic diseases, and familial, including family situation and seeking care/attention. As Brofenbrenner’s bioecological model demonstrated, these determinants are part of the microsystem involving individuals, family, friends, school, and neighbourhood.

Dieting and binging appeared to be the most experienced DEBs followed by induced vomiting. Most adolescent girls confirmed that they are not content with their weight and shape. Similarly, most of them believed that girls their age are not happy with their weight and appearance, and would rather lose weight. Most of them perceived that the frequency of DEBs would be less with age, as they explained that the level of comprehension and consciousness would increase with age, improving the ability to control emotions and actions and develop better recognition in addressing risk and detrimental effects associated with such behaviours. Contrary to our previous finding, age is a predictor of DEB with older students reporting a higher risk compared to younger students [18]. Likewise, the results of the current study do not align with our systematic review finding that identified a higher risk of DEB reported among older students compared to younger students [53]. It is possible that the elevated frequency of DEBs with age would be due to increased self-consciousness and awareness, which might aggravate concerns related to body image and social acceptance [54].

Most adolescent girls identified that negative cognitions, conscious imitation/copying behaviours, bullying, comparisons, and negative comments are the most common determinants of DEBs. Additionally, some adolescents mentioned that biological and family determinants can play a role in increasing the risk of DEBs.

Identifying discussed determinants through Brofenbrenner’s bioecological model, the microsystem, which consists of the individual and the immediate environment—including family, friends, school peers, and neighbourhood—is the primary system encompassing the majority of DEB determinants [38,39,40]. It is essential to emphasise the role of the techno-subsystem surrounding the individual, with the continuous use of devices and phones shaping and modifying concepts/perceptions on weight and redefining the standards of beauty and attractiveness [42,43]. The connections of the meso-system, mass media in the exosystem, and social contexts in the macrosystem provoke the phenomenology of DEBs by increasing the pressure of having a thin and slender body supported by mass media. This creates a framework of dieting and body dissatisfaction, which could be defined as a preliminary stage for DEBs and eating disorders [55].

Negative cognitions were primarily focused on confidence issues and negative body image. These cognitions, according to the girls, are associated with emotions of depression, anxiety, and an immediate urge to change their appearance and/or lose weight. The role of negative cognitions and self-beliefs has been confirmed in the literature as a causal factor leading to eating disorders [9,56]. Mediational analyses on adolescents demonstrated that negative self-image and body dissatisfaction have both direct and indirect influences on DEBs through self-esteem and aversive emotional status [57]. A group of US eighth- and ninth-grade adolescent girls completed measures related to self-cognitions and self-esteem. Based on self-schema, the cognitive framework of self-beliefs and self-information that directs an individual’s perceptions, adolescent females who exhibit high levels of eating disorder symptoms experience more negative self-beliefs and schemas. [58,59]. DEBs are influenced by cognitive schemas, low confidence, and skewed information processing, which creates a demand for control and an idealised desire for body shape and weight [60].

Conscious imitation and copying behaviours emerge as a result of admiration and preference of other individuals’ appearance or shape. Adolescent girls explained that this can either happen in real-life by imitating the behaviours of friends/schoolmates or on social media by copying celebrities’ actions on Instagram or TikTok [23,24,25]. They intentionally mimic their behaviours and copy their actions to resemble their body image and to reach a comparable physique. Furthermore, they emphasised that severe methods of exercising and dieting practised by Korean celebrities are imitated. Korean celebrities or K-pop stars, as called by adolescents, have large popularity among female adolescents in Saudi Arabia, notably among intermediate school students, aged 12–14 years. As they are captivated by their bodies and shapes, they follow them on apps and start copying their strict dieting behaviours to lose weight. Recent systematic reviews have indicated that adolescents’ usage of social media increases their vulnerability to DEB by contributing to body dissatisfaction and low self-esteem [61,62]. The dilemma that adolescents experience when imitating or copying a behaviour stems from feelings of inferiority and inadequacy, since they do not measure up to their models, associated with fear of social rejection or dislike [63].

Bullying among school-age children and adolescents is an ongoing phenomenon that has substantial short- and long-term effects on mental and physical health [64]. Adolescent girls identified bullying—either through traditional bullying methods or cyberbullying—as a common determinant of DEBs. It can be detrimental, leading to severe consequences due to its direct association with suicidal thoughts and depressive/hopeless moods, with sadness serving as a mediating factor between bullying and suicidal thoughts [65]. Compared to their peers of normal weight, adolescents who are overweight and obese have a higher likelihood of experiencing bullying [66]. A study that explored the impact of bullying and DEBs on Australian adolescents aged 11–19 years concluded that bullying concerned with weight and shape increased the frequency of DEB, particularly induced vomiting [67]. Current research has indicated a positive association between bullying and risk of DEBs, with bullied and teased adolescents having higher susceptibility to DEBs and negative body image compared to adolescents who are not bullied [8,68]. Due to technology and social media, adolescents are increasingly experiencing cyberbullying, which has been linked to symptoms of eating disorders [69].

Comparisons and negative comment sources were addressed by adolescent girls as either intrapersonal (originating from oneself) or interpersonal (originating from parents, siblings, relatives, and friends). Intrapersonal comments are focused on internal cognitive criticism of weight and/or shape and negative self-talk related to body image. Findings suggest that self-criticism and low self-differentiation are strong and significant determinants of disordered symptoms [70,71]. Interpersonal comparisons and comments are negative verbal messages on weight or appearance delivered by others. Interpersonal comparisons and comments from parents were defined by girls as the most destructive. Parental comments have a detrimental effect on adolescents by directly contributing to distorted body image and significantly causing DEBs [72,73]. A study explored the association of parental comments with DEBs among Asian young adults aged 18 to 25 years and confirmed that parental comments on weight and shape can have a substantial influence on increasing body dissatisfaction and DEBs [74].

Some students mentioned biological determinants, including puberty and hormonal changes and fear of chronic diseases, as potential determinants of DEB. Among the most commonly cited risk phases for the emergence of eating issues is puberty, suggesting that psychological influences—body dissatisfaction and low confidence—can be significant mediating sources of risk in this sensitive stage [7,8,75]. Early pubertal timing increases this risk of DE and its symptoms, as it is related to a higher risk of dieting [76]. This is explained due to hormonal shifts during puberty; progesterone and estradiol are secreted in females, and increases in the amounts of steroids in circulation have a direct impact on psychological and behavioural characteristics, such as anxiety and eating habits [77]. Research has indicated the role of reproductive hormones on the symptoms of eating disorders; however, the effect elucidated in the literature is insufficient, and longitudinal research is required to measure changes in these hormones and their pathogenic mechanism [78]. Adolescent girls described fear of chronic diseases, particularly weight-related diseases, mainly diabetes and hypertension. No literature has explored the fear of chronic disease as a determinant of DEB; however, fear of weight gain is defined as the core of eating disorders [79].

Family-related determinants were mentioned by adolescent girls as possible causes leading to DEBs. Family situation and other family-related factors can play a role in the development of DEBs [20,21,80]. A systematic review compared family functioning in families with eating disorders and control families and found families with EDs to be less functional [81]. A positive family meal atmosphere was associated with a lower likelihood of DEBs in adolescents [82]. Adolescent girls also stated that they could engage in DEBs to seek attention and care from their parents.

Determinants leading to DEBs are complex, originating from a reciprocal interaction with biological, psychological, and societal causes [6]. Currently, there is a dearth of empirical research on the prevalence of determinants of DEBs among adolescents, ultimately leading to classify this study as a novel piece of research for understanding DEB and its main determinants. To clearly grasp the causal relationship and identify the primary and mediating components, it is necessary to address determinants across a wider spectrum. DEBs appeared to be common in adolescents attending intermediate and secondary schools in Riyadh, Saudi Arabia [18,53]. Comprehending determinants that cause DEBs would be immensely beneficial to minimise risk and to create primary preventive approaches appropriate for school settings to limit such behaviours.

## 5. Limitations

A significant proportion of older adolescent girls, for whom consent was given, declined to participate. Specifically, more than half of students from the 128 secondary school declined to take part. The low response rate may in part be explained by the adolescents’ tendency to be shy and avoid discussing personal experiences. However, it might also be explained by the parents’ concerns and anxiety about the wellbeing of their daughters. Despite the low response rate and given the qualitative nature of the study, we are confident that saturation, the point at which no more major nuances or insights can be found, can be achieved with 16 to 24 interviews [83]. The students participated differently in the interviews, which could be due to social acceptability biases, as some may convey information that is regarded to be socially acceptable [84]. Although focus groups may be the optimal method for data collection in other topic areas, individual interviews were deliberately chosen to facilitate greater transparency regarding thoughts and emotions on body image and personal experiences with DEBs [85]. The data collected might not be representative of the population of adolescents. A more diverse sample might increase the accuracy and diversity of responses.

## 6. Conclusions

This study, to the authors’ knowledge, is the first study exploring the determinants of DEBs among adolescent girls in intermediate and secondary schools in Riyadh, Saudi Arabia. The interview process included 18 students with a high risk of DEB. Negative cognitions, conscious imitation/copying behaviours, bullying, comparisons, and negative comments, as well as biological and family determinants, are the determinants identified by adolescent girls as causing DEB. This study can be considered as an initial guide, providing a framework to increase understanding of the determinants of DEB in adolescent girls in Saudi Arabia. The findings could be used to inform the development of interventions to address underlying causes of DEB in schools and other relevant settings.

## Figures and Tables

**Table 1 nutrients-16-02119-t001:** Themes/sub-themes and Bronfenbrenner’s bioecological model systems.

Themes	Sub-Themes	Codes	Bioecological Model System
Negative cognitions	Self-esteem and confidence issues negative body image	1a 1b	Microsystem—individual
Conscious imitation/copying behaviours	Real-life models Social media—TikTok and Instagram	2a 2b	Microsystem and techno-subsystem—individual
Bullying	Traditional bullying- Siblings and school peers Cyberbullying	3a 3b	Microsystem and techno-subsystem—individual, family, school, friends, and neighbourhood
Comparisons and negative comments	Intrapersonal comparisons Interpersonal comparisons and comments	4a 4b	Microsystem—individual, family, school, friends, and neighbourhood
Biological determinants	Puberty and hormonal changes Fear of chronic diseases	5a 5b	Microsystem—individual
Family-related determinants	Family situation/house environment Care and attention	6a 6b	Microsystem—individual and family

## Data Availability

For the purpose of open access, the author has applied a Creative Common Attribution (CC BY) licence to any Author Accepted Manuscript version arising. All data generated or analysed during this study are included in this published article.

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
