# Peer review of "Determinants of Disordered Eating Behaviours (DEBs) among Adolescent Female School Students in Riyadh, Saudi Arabia: A Qualitative Study"

_nutrients, 2024, doi:10.3390/nu16132119_

Round 1
Reviewer 1 Report
Comments and Suggestions for Authors
I have read with interest the article entitled:
Determinants of Disordered Eating Behaviours (DEBs) Among Adolescent Female School Students in Riyadh, Saudi Arabia: A Qualitative Study ». The title which describes the article and the abstract as well as the introduction and the objectives are clearly laid out. Τhe methodology is well described. The results are clearly led out and in a logical sequence. The conclusions are well discussed. Nevertheless, there are some serious limitations:
- More than half of students from the 128 secondary 649 school declined to take part, that is the response rate is less than 50% .This extremely low response rate may introduce serious systematic errors. What are the possible causes for such a law response rate? Please explain/.
- Small sample size: That affect the genaralizability of the results. Also, It might have been useful to use text analytics in a research of this type, but such a small sample does not allow the use of ths method-
Reviewer 2 Report
Comments and Suggestions for Authors
This is a necessary study on a subject on which much crucial information is unknown, and so this research contributes to filling an essential scientific gap. However, this study needs some improvements, in my opinion, which I will explain below:
- Much of the literature presented is too old. The sources on which the study is based should be updated.
- The study states that there is a need to know how an DEB originates and develops in Saudi adolescent girls, but does not clearly explain why it is necessary to consider them differently from adolescent girls in other cultures. The authors should clearly explain what cultural circumstances might lead to differences in the emergence and development of LBP for a Saudi adolescent girl.
- The discussion names studies other than those presented in the theoretical framework of the study, which is a mistake. If the studies are important and therefore appear in the discussion, at least some of them should appear in the theoretical introduction.
- As the DEBs have an important social component and this is a qualitative study, the absence of focus groups is not understandable. They should be included or appear as one of the main limitations of the study.
